

# A novel traffic optimization method using GRU based deep neural network for the IoV system

Wu Wen[1], Dongliang Xu[1] and Yang Xia[2]

[1] ChongQing Technology And Business Institute, ChongQing, China
[2] ChongQing Open University, ChongQing, China

## ABSTRACT

At present, China is moving towards the direction of "Industry 4.0". The development of the automobile industry, especially intelligent automobiles, is in full swing, which brings great convenience to people's life and travel. However, at the same time, urban traffic pressure is also increasingly prominent, and the situation of traffic congestion and traffic safety is not optimistic. In this context, the Internet of Vehicles (also known as "IoV") opens up a new way to relieve urban traffic pressure. Therefore, in order to further optimize the road network traffic conditions in the IoV environment, this research focuses on the traffic flow prediction algorithm on the basis of deep learning to enhance traffic efficiency and safety. First, the study investigates the short-time traffic flow prediction by combining the characteristics of the IoV environment. To address the issues that existing algorithms cannot automatically extract data features and the model expression capability is weak, the study chooses to build a deep neural network using GRU model in deep learning for short-time traffic flow prediction, thereby improving the prediction accuracy of algorithm. Secondly, a fine-grained traffic flow statistics approach suitable for the IoV situation is suggested in accordance with the deep learning model that was built. The algorithm sends the vehicle characteristic data obtained through GRU model training into the fine-grained traffic flow statistics algorithm, so as to realize the statistics of traffic information of various types of vehicles. The advantage of this algorithm is that it can well count the traffic flow of multiple lanes, so as to better predict the current traffic status and achieve traffic optimization. Finally, the IoV environment is constructed to confirm the effectiveness of the prediction model. The prediction results prove that the new algorithm has good performance in traffic flow statistics in different scenarios.

## INTRODUCTION

As the country's economy and people's quality of life have continued to rise, cars have become increasingly popular, and been an essential element of daily life. It is undeniable that increasingly intelligent cars bring great convenience to people's production life, but at the same time, it also makes the urban traffic management more and more challenging.

Corresponding author
Wu Wen, kevin983913@163.com

For instance, traffic congestion is getting worse and worse, which can easily lead to traffic accidents, energy consumption and environmental pollution and other social problems. Especially during rush hour or tourist holidays, with the increase of traffic flow on the road, traffic congestion and traffic accidents occur frequently, causing a vicious cycle. It not only seriously affects people's travel experience and quality of life, but also causes great damage to people's lives and properties. Therefore, it is crucial to find a quick solution to the traffic congestion issue. In recent years, the Internet of Vehicles technology has gained widespread recognition and is now thought of as a fresh approach to solving traffic issues in the future thanks to the quick application of new technologies including the Internet of Things, 5G technology, and environmental perception (*Manias & Shami, 2021*).

As a representative application of the Internet of Things in the field of transportation, the IoV uses various electronic sensors in the car, and utilizes the latest communication technology to establish a powerful and efficient service information network, so as to realize the interconnection between vehicles and all things (such as vehicles, people and roads, etc.) (*Qureshi et al., 2021*). In other words, in the IoV environment, data can be exchanged and shared between vehicles and vehicles, vehicles and people or vehicles and roads in real time, and they can also work closely with the road traffic control system to provide data support for traffic decision making and optimization. As a result, IoV becomes an indispensable part of building modern intelligent transportation systems (*Chiroma et al., 2021*). It contributes to a reduction in traffic congestion and accidents as well as an increase in the capacity of road networks for transportation, ensuring that people can travel safely. There are numerous causes of traffic congestion today. It cannot simply rely on the support of advanced technologies such as Internet of Vehicles, but more importantly, it would predict the trend of traffic flow changes that will occur in the near future. It can be argued that the enormous amount of data information in the IoV offers strong data support for the studies linked to traffic flow prediction (*Zhang, Lu & Li, 2020*). In the IoV environment, traffic flow prediction can assist relevant traffic management departments to grasp the traffic status in time, formulate effective traffic flow management plans, and dynamically control traffic flow, so as to optimize traffic resources and improve the traffic environment. On the other hand, it can offer suitable travel routes in advance so that people can avoid congested areas or times, saving travel time. Moreover, with the development demand of intelligent transportation, traffic flow statistics in the IoV environment has become an important issue in computer vision (*Zhao et al., 2020*). Meanwhile, fast and effective traffic flow statistics can well relieve traffic congestion and reduce the possibility of traffic accidents, thus reducing the pressure of urban traffic. Therefore, designing a reliable, efficient and accurate traffic flow prediction model is of very positive significance for optimizing the traffic of IoV road network.

In traffic flow prediction research, the scale of data involved is often very large. The benefit of deep learning technology is simply that it can extract useful information from vast amounts of data, which aids in understanding the changing traffic laws more quickly and precisely and considerably enhances the efficiency and accuracy of traffic flow prediction. Therefore, it is very necessary to research and develop road network traffic optimization based on deep learning in the IoV environment. Different from traditional machine

learning methods, deep learning methods rely on data for representation learning, which can directly train models on a large amount of original data, thereby extracting abstract and high-level feature representations (*Montieri et al., 2021*). With the advancement of theories relating to deep learning , many typical deep neural networks have appeared, such as convolutional neural network (CNN) (*Park & Oh, 2021*), recurrent neural network (RNN) (*Lalapura, Amudha & Satheesh, 2021*) and various improved structural networks. All of them have achieved excellent performance in traffic flow prediction research. *Jie et al. (2022)* proposed a state space neural network. This method utilized the spatial dimension characteristics of traffic flow data and achieved ideal prediction results. *Kouziokas (2020)* proposed an long short-term memory (LSTM)-based algorithm for traffic flow data prediction. The algorithm took full advantage of the features of LSTM to better extract the temporal features of traffic flow data, and then achieved the prediction of traffic flow data. In *Li et al. (2021)*, an improved neural network combining CNN and LSTM was studied and used for traffic flow prediction. In *Boukerche & Wang (2020)*, a road network traffic flow prediction model architecture was constructed using stacked RNN units. The literature *Jamiya & Rani (2021)* and *Ali et al. (2021)* investigated the application of deep neural networks (including gated recurrent unit (GRU), LSTM, and RNN) and traditional machine learning methods (support vector machine (SVM), autoregressive integrated moving average (ARIMA), sparse autoencoder (SAE), radial basis function (RBF)) in traffic flow prediction, respectively, and compared the prediction performance of these representative models in the analysis. *Ren et al. (2021)* integrated the advantages of both LSTM and RNN models in traffic flow prediction and designs a spatio-temporal recurrent convolutional neural network model.

The above research demonstrate that the deep learning algorithms, like RNN, LSTM and GRU, are being extensively utilized in the field of traffic management. However, as China's urban traffic road network becomes more and more complex, the traffic operation conditions change rapidly, which makes higher demands on the timeliness and accuracy of traffic flow prediction. The characteristics of prediction accuracy and stability must be thoroughly taken into account when analyzing the prediction model. As a result, this research introduces a traffic optimization method based on GRU model combined with a fine-grained traffic flow statistical calculation method for the Internet of Vehicles. The purpose is to increase the generalization ability and prediction accuracy of existing prediction methods. The main research contents are as follows: (1) This research decides to utilize the GRU model to build a deep neural network in accordance with the characteristics of traffic flow data to address the issue that the present algorithms cannot automatically extract data features and the model expression ability is weak. In this manner, it is possible to predict traffic flow in the IoV environment over the short term, and algorithm prediction accuracy is improved. (2) In order to realize the traffic information statistics of different vehicle types, a fine-grained traffic flow statistics algorithm suitable for the IoV scenario is proposed for the deep learning model built. This is because the traditional traffic flow statistics algorithm is particularly prone to the issue of missed detection or false detection. (3) A IoV environment is constructed to demonstrate the effectiveness of the prediction model. The prediction results prove that the new algorithm has good performance of traffic

statistics in different scenarios, which fully reflects the significant advantage of the new algorithm in terms of stability and accuracy of prediction.

## RELATED WORK

### Deep learning

As one of the branches of machine learning, the concept of deep learning can be traced back to 2006, when it was introduced by the scholar G. E. Hinton. Unlike traditional machine learning methods such as SVM (*Aurangzeb, Ayub & Alhussein, 2021*), Boosting (*Konstantinov & Utkin, 2021*), and SAE (*Ji, Zhang & Wu, 2020*), deep learning utilizes a deep network architecture containing multiple layers to perform some processing on the input data in turn, while initializing the neural network parameters using unsupervised pre-training methods. By training the model iteratively, the layers of the network become deeper and deeper, and the original data are trained to different degrees, gradually making the training results meet the feature representation required for the target task. Finally, the extracted data features are classified and regressed to produce the results (*Naghizadeh, Metaxas & Liu, 2021*). The deep learning can directly process the original data at multiple levels, effectively avoiding the process of preprocessing complex data with traditional machine learning models. Thus, manual intervention is reduced, and its prediction accuracy and operation efficiency are greatly improved. The core component of deep learning models is a large number of neuron structures that hierarchically perform feature extraction on original data through multiple transformation stages and interpret these data (*Ke et al., 2021*). Figure 1 displays the data processing flow.

### Neurons

The most fundamental components of a deep neural network are neurons (*Bodyanskiy & Antonenko, 2021*). Numerous neurons can be combined to form a complicated deep neural network. They are connected according to certain rules and then distributed in each layer of the network. In general, neurons may consist of either linear products or activation functions using nonlinear parts. Figure 2 depicts its structure.

Usually, each neuron typically does not exist in isolation in deep neural network models. On the other hand, they simultaneously connect to a number of other neurons in different network layers. In this way, the output signals sent from other neurons can be used as input signals for that neuron. Suppose the input signal of each neuron is defined as $S = (s_1, s_2, ...., s_m,)$, and its corresponding weight matrix is $V = (v_1, v_2, ...., v_m)$. Then the output signal of this neuron can be expressed by Eq. (1).

$$z = f\left(\sum_{i=1}^{m} v_i s_i + a\right) \tag{1}$$

where $f$ represents the activation function and $a$ means the bias vector. $v$ and $a$ are the key variables of the neuron and will significantly affect the output of the neuron. During the model training process, these two variables are continuously adjusted to achieve model optimization. The activation function $f$ also plays a pivotal role in the neuron's processing of the input data and is one of the key aspects of the model training. If the neuron consists

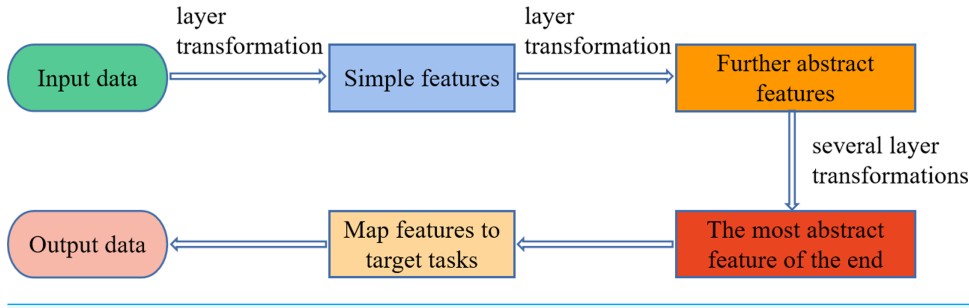

**Figure 1** **Flow chart of data processing for deep learning.**

of only linear functions, then the output results obtained through model training are only linear transformations of the input data, which greatly limits the learning ability of the model. Therefore, the nonlinear part of the activation function $f$ is introduced into the neurons, so as to enhance the nonlinear mapping learning ability of deep neural networks. Since the characteristics of different activation functions are also different, their impact on the model training results is not exactly the same. Currently, the commonly used activation functions are *sigmoid* function and *tanh* function. The former has the advantage of being easy to derive and the disadvantage of having problems such as gradient disappearance. The latter has the advantage of faster convergence.

## Deep feedforward network

The deep feedforward network is one of the most well-known artificial neural networks, which has a typical deep neural network structure (*Kozlenko, Zamikhovska & Zamikhovskyi, 2021*). It is usually a one-way multilayer network structure with no signal feedback consisting of input, hidden and output layers, so its whole network can be expressed as a directed acyclic graph. Moreover, the network contains a large number of multilayer neurons. The data is fed into the network's previous layer's neurons for processing, and after the processing is finished, the output is fed into the network's next layer's neurons. Figure 3 depicts the structure of the deep feedforward network.

In Fig. 3, the deep feed-forward neural network uses forward propagation to calculate the output of the model. The input layer is located in the first layer of the entire network model and takes on the role of receiving external data signals. The hidden layer usually contains numerous multilayer neurons. Moreover, the fitting degree of the model is often related to the number of layers in the hidden layer. The more layers, the better the fitting degree of the model, which is more conducive to solving complex problems. Meanwhile, it is important to note that if the network has too many hidden layers, it will lead to an increase of neuron parameter variables in the network, which will increase the model training burden and cause problems such as gradient stability and network degradation. The output layer is at the end of the network structure, whose main task is to output the training results of the model.

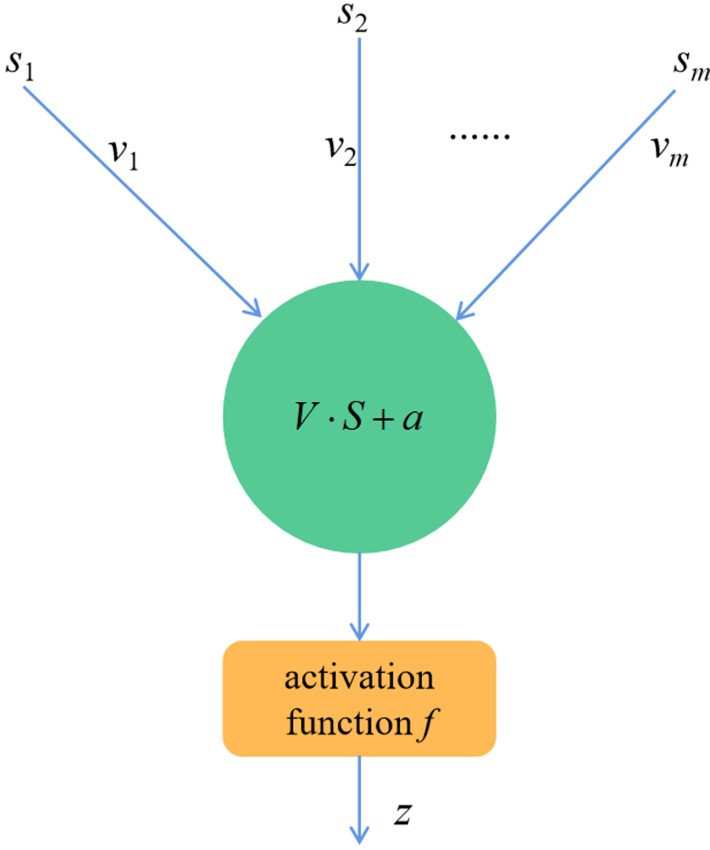

**Figure 2** Neuronal structure.

### Error back propagation

The artificial neural network's scale is substantially increased by the addition of multi-level hidden layers, which also improve the model's capacity for feature representation. To ensure that the final output results match the feature representation required by the target task, it is important to note that the network's parameters must be adjusted repeatedly throughout the model training process. However, as the network model's layers get deeper and deeper, this could also strain the model and reduce its effectiveness. One of the frequently utilized algorithms for training neural networks at the present is the error back propagation algorithm. It can effectively reduce the number of solution parameter gradient expressions, so as to quickly adjust the network model parameters. Figure 4 depicts its training process.

As illustrated in Fig. 4, the output data from the forward propagation in the neural network serves as the input data in the error back propagation algorithm. It uses the loss function to calculate the error of the model prediction, and then the error signal is backward propagated through the network layer by layer. It also utilizes the chain rule to calculate the gradient of the weights of each neuron and updates the weights layer by layer using the gradient descent algorithm.

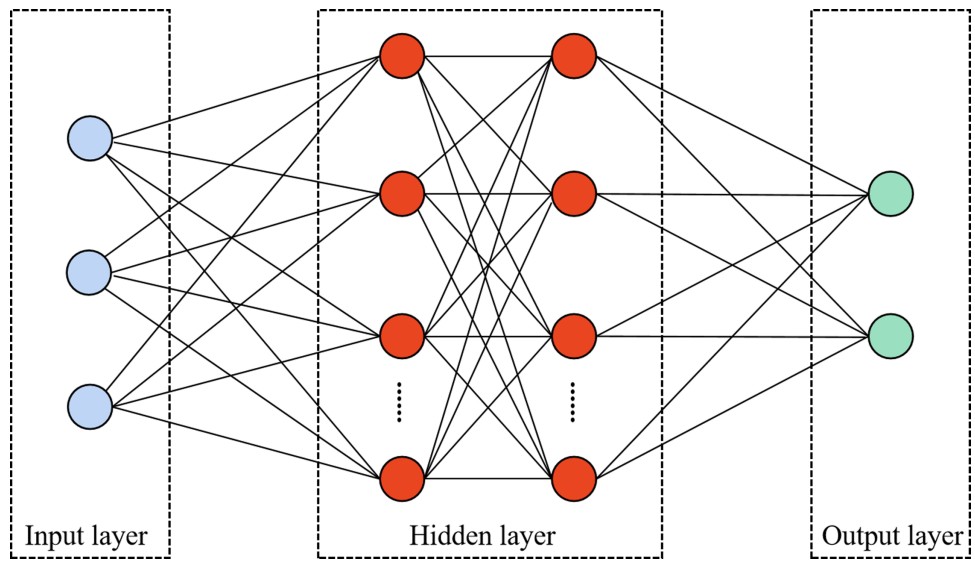

**Figure 3** The structure of the deep feed-forward network.

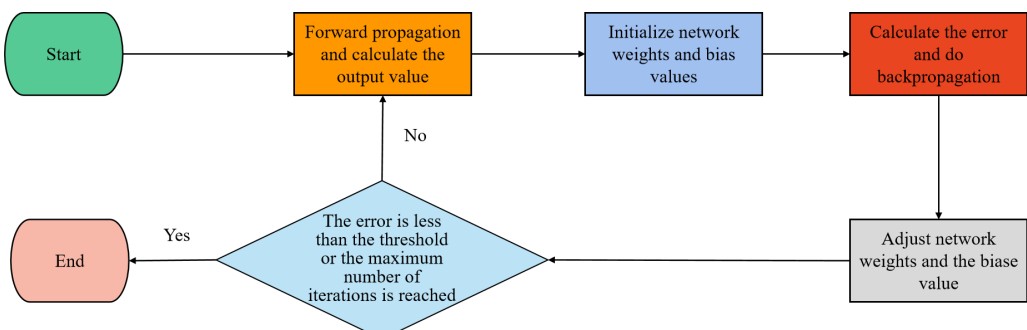

**Figure 4** The training process of backward propagation.

## Deep learning-based traffic flow prediction model

This study optimizes the GRU neural network model starting from the direction of road network traffic optimization in the IoV environment. In this manner, the generalizability and prediction accuracy of current traffic flow prediction techniques will be increased. The prediction model mainly includes the steps of traffic data collection, prediction model training, result analysis and traffic alert. Figure 5 displays the detailed workflow of the model.

Figure 5 shows the detailed workflow of the model. (1) Traffic data collection. Traffic data such as weather, vehicle type, number of vehicles, and so on are collected in real time using road measurement sensors, cameras, and other related devices in the IoV environment. (2) Model training. The collected traffic data are pre-processed and then

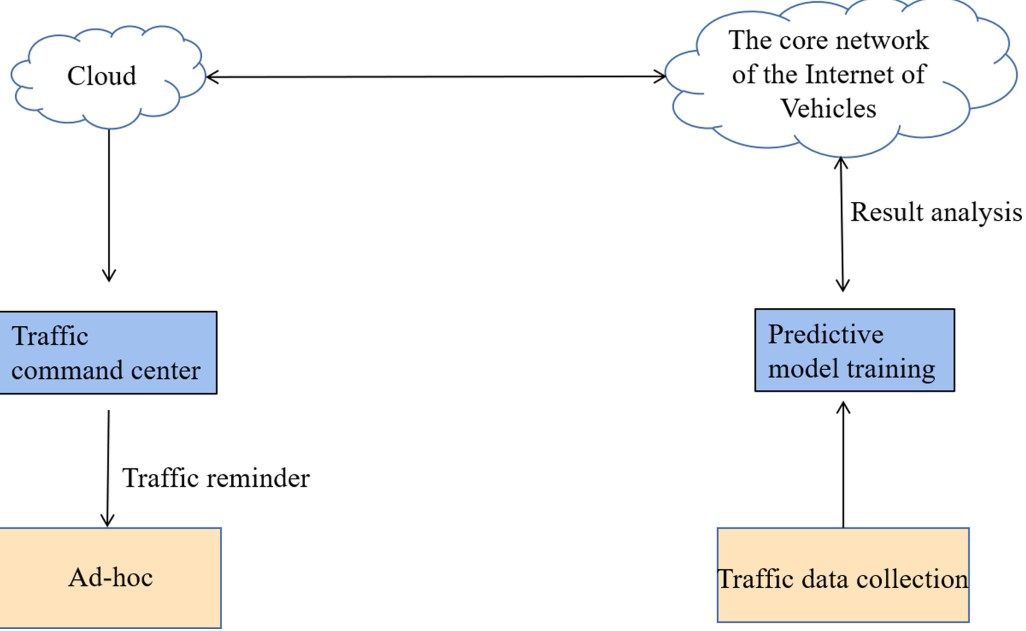

**Figure 5  Flow chart of traffic flow prediction based on deep learning for IoV.** The detailed workflow of the model.

directly used as input data for the prediction model for model training. (3) Results analysis. The prediction results of the model are transmitted to the traffic command center using advanced network communication technology. (4) Traffic alert. The traffic command center further analyzes and calculates the prediction results in depth and feeds the results to the relevant vehicle units, thus providing reasonable suggestions for people's travel arrangements. The purpose of road traffic optimization is finally realized.

## GRU module design

The article chooses to use the GRU (gated recurrent unit) model to build a deep neural network for short-time prediction of traffic flow in the IoV environment. The GRU neural network is a model improved on the basis of LSTM by Cho et al. (*Behera, Misra & Sillitti, 2021*). Its performance is close to that of LSTM, but the difference is that it combines the forgetting gate and the input gate in LSTM as a reset gate, so the internal structure of the GRU model is simpler.

A simple GRU model typically only has two gate structures: the update gate and the reset gate. The GRU model has one fewer gate structure than the LSTM model, which results in a decrease in the total number of parameters for the overall model. Figure 6 depicts the internal structure of the GRU model.

In Fig. 6, the two gate structures of the GRU model are update gate $n_t$ and reset gate $s_t$. The $n_t$ is mainly used to determine the degree of receiving information from the previous moment, and the $s_t$ is mainly utilized to determine the degree of ignoring information from the previous moment. $n_t$ has a larger value indicating a greater degree of receiving information from the previous moment cell, and $s_t$ has a larger value indicating that the

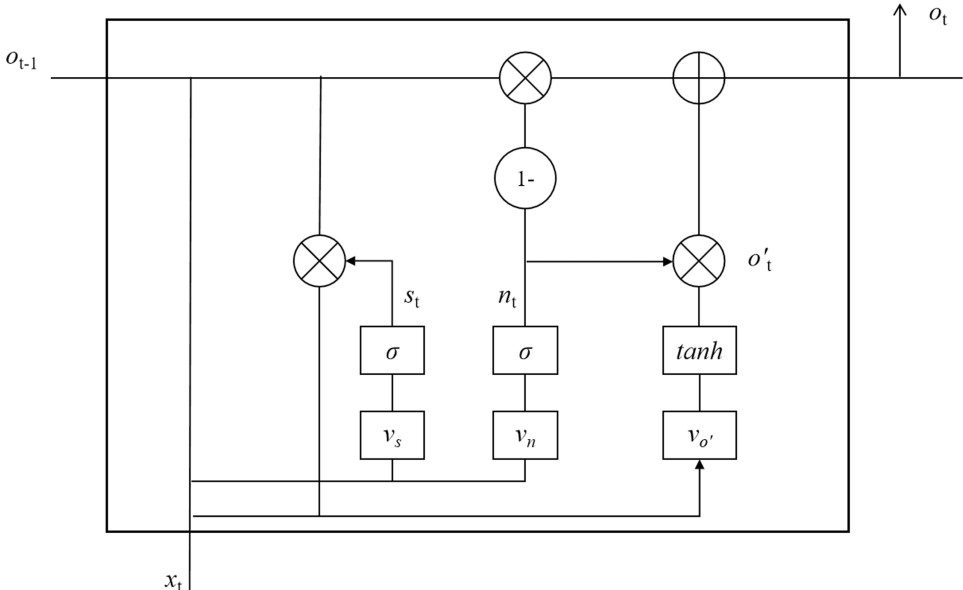

**Figure 6 Internal structure of GRU model.**

state information of the previous moment cell has less influence on the state of the current cell. The expressions of both are shown in Eq. (2) and Eq. (3), respectively.

$$n_t = \sigma(v_n \cdot [o_{t-1}, x_t]) \tag{2}$$

$$s_t = \sigma(v_s \cdot [o_{t-1}, x_t]) \tag{3}$$

where $v_n$, $v_s$ denote the weight matrix of the two gates respectively, $x_t$ denotes the current cell input, $o_{t-1}$ denotes the state information of the cell at the previous moment. $o_t$ denotes the output value of the cell at the current moment, which is defined in Eq. (4) and Eq. (5).

$$o_t = (1 - n_t) \cdot o_{t-1} + n_t \cdot o_t^{'} \tag{4}$$

$$o_t^{'} = \tanh(v_{o'} \cdot [s_t \cdot o_{t-1}, x_t]) \tag{5}$$

Since there is one less output gate in the GRU model to control its output, the state information of its cells at all moments is output to the cells at the next moment. This makes GRU eliminate the operation process of multiplying several matrices, which makes the GRU model more advantageous in solving the short-time traffic flow prediction problem. This is exactly the reason why the GRU model is chosen in this article.

## Fine-grained traffic flow statistical calculation method

Since the traditional traffic flow statistical calculation method is prone to the problem of missing or wrong detection, a fine-grained traffic flow statistical calculation method is proposed for the constructed deep learning model to better calculate the traffic flow

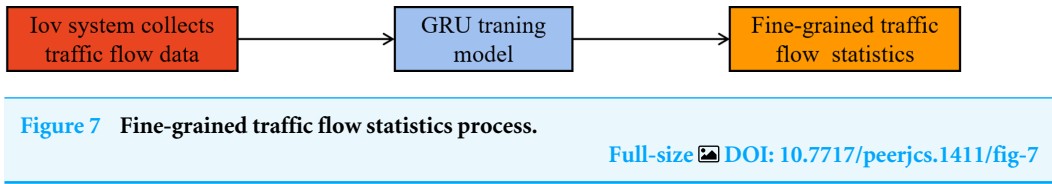

**Figure 7   Fine-grained traffic flow statistics process.**

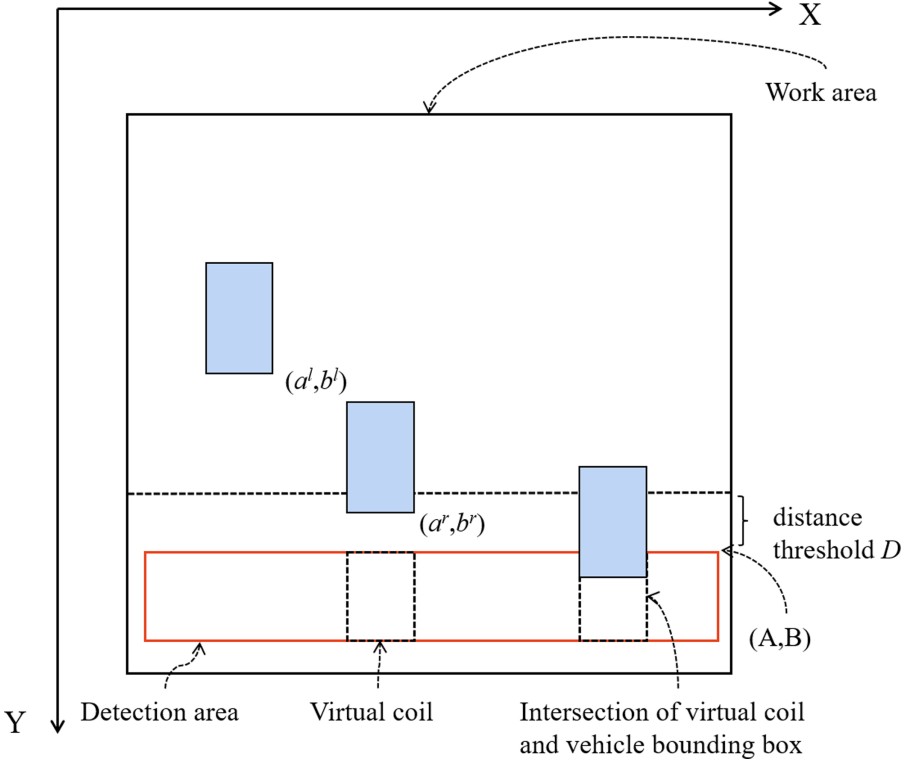

**Figure 8   Working principle of algorithm.**

data in the IoV scenario. In this study, the road traffic data collected by roadside sensors and cameras are passed to the GRU prediction model for model training, and the vehicle location and category information of each vehicle is obtained through the model training, and then the extracted information of each vehicle is fed into the fine-grained traffic flow statistical algorithm to realize multi-lane and multi-type traffic flow information statistics. Figure 7 displays the whole flow chart.

It is important to note that the work area must be set for this method. The only vehicles that will be counted are those that appear in the work area,any other vehicles are simply ignored. Figure 8 displays the algorithm's working schematic.

It is assumed that the bounding box information of the vehicle is determined by the coordinate values of the upper left and lower right corners, which are defined as $(a^l, b^l)$ and $(a^r, b^r)$, respectively. $(A, B)$ represent the coordinates of the upper right corner of the detection area. The upper left coordinate of the virtual coil is$(a^u, b^u)$ and the lower right coordinate is $(a^v, b^v)$. $t$ represents the logical time when the virtual coil is created. The

expression of the virtual coil is shown in Eq. (6).

$$R = (a_1^u, b_1^u, a_1^v, b_1^v, t_1), \ldots, (a_n^u, b_n^u, a_n^v, b_n^v, t_n) \tag{6}$$

If the bounding box information of the vehicle can satisfy both the conditions of Eq. (7) and Eq. (8), a virtual coil with that vehicle is established in the detection area.

$$B - b^r < D \tag{7}$$

$$B - b^r > 0 \tag{8}$$

where $D$ denotes the distance threshold between the bounding box of vehicle and the detection area. When $|B - b^r| < D$ is used, each frame needs to determine whether there is a virtual coil corresponding to the bounding box of the current vehicle, whose expression is shown in Eq. (9).

$$\frac{\max(a^v, a^r) - \min(a^u, a^l)}{a^v - a^u} > L \tag{9}$$

where $L$ denotes the threshold value and $L \in [0, 1]$. The above formula indicates that after the current vehicle width is projected onto the virtual coil width, only when the ratio of the projected value to the virtual coil width is greater than the threshold $L$, it is determined that the current vehicle has a virtual coil corresponding to it. When the bounding box of the vehicle enters into the virtual coil, it is necessary to execute the vehicle counting process at this time, and when the condition of Eq. (10) is satisfied, the vehicle is counted and the current virtual coil is deleted.

$$\frac{\text{area}(F \cap G)}{\text{area}(G)} > H \tag{10}$$

where $F$ represents the rectangular bounding box of the vehicle to be judged, $G$ represents the virtual coil frame to be judged, and $H$ is the vehicle count threshold. At the end, the following situation should also be considered to ensure the robustness of this algorithm, that is, if a certain vehicle creates a certain virtual coil, but for some special reasons, this virtual coil is not deleted after the vehicle passes. In this case, the virtual coil will always exist and may have a negative impact on the later statistics. Therefore, such coils need to be cleared and the virtual coils are deleted when the conditions such as Eq. (11) are met.

$$t_{\text{now}} - t_{\text{create}} > T_{\text{max}} \tag{11}$$

where $T_{max}$ is the longest logical time that the virtual coil can be tolerated to exist.

## Simulation experiments

The research also performs simulation experiments to confirm the application performance of the traffic flow prediction model designed in this work in the IoV environment. To test its effectiveness, four different scenarios of road videos in varied weather conditions—sunny, cloudy, rainy, and evening—will be constructed using the traffic videos that the camera has captured.

## Experimental data

The research in this article focuses on predicting the traffic flow trends of a regional traffic road in the near future based on the traffic data collected in the IoV environment. Therefore, the traffic dataset in this experiment is mainly derived from a hybrid dataset composed of manually labeled dataset and UA-DETRAC public dataset.

The manually labeled dataset contains a total of 1,618 traffic images with a size of 540 × 960, and the shooting scenes involve four situations: sunny, cloudy, rainy and evening. Some of these images were collected by using a camera on a flyover in a certain city, and the other part was obtained by web crawlers. The dataset includes the front and back sides of vehicles. The vehicle types include small cars, buses, cabs, trucks, SUV, and pickup trucks. As a commonly used public vehicle detection dataset, the images within the UA-DETRAC (*Perreault et al., 2021*) dataset are all collected on the flyover. The dataset collected a total of 82,118 images of 8,250 vehicles in cloudy, evening, sunny, and rainy conditions. The size of each image is 540 × 960 and it involves thirteen types of vehicles, including cabs, cars, SUV, trucks, commercial vehicles and so on. However, since the number of images for each type of vehicles is not equally distributed in the UA-DETRAC dataset, the number of police cars and flatbed trucks is relatively small. In view of this, the article selects the images of nine types of vehicles, including SUV, Sedan, Taxi, Van, MiniVan, Truck-Box, Bus, Truck-Util, and Pickup, from the UA-DETRAC dataset, totaling 11,618. Then the hybrid dataset was formed with 1,618 manually labeled images to ensure the adequacy of computational resources. Then the hybrid dataset was divided into four conditions: sunny day, cloudy day, rainy day, and evening for performance testing. Among them, 10,588 images were selected from the hybrid dataset as the training dataset, and the remaining images were used as the test dataset.

## Evaluation indicators

In order to evaluate the performance of the traffic prediction model, this article uses the accuracy rate as the evaluation index. It is defined in Eq. (12).

$$\text{Accuracy} = \begin{cases} 1 - \dfrac{N_{\text{error}}}{N_{\text{true}}}, \text{if } N_{\text{error}} < N_{\text{true}} \\ 0, \text{otherwise} \end{cases} \tag{12}$$

It is worth noting that in Eq. (12), if the number of erroneous vehicles exceeds the actual number of vehicles, it exceeds the upper error limit designed in this article. In this case, the prediction accuracy of the model is judged to be 0. Where $N_{true}$ indicates the number of real traffic flow vehicles and $N_{error}$ means the number of erroneous vehicles, which is defined in Eq. (13).

$$N_{\text{error}} = |N_{\text{true}} - N_{\text{count}}| \tag{13}$$

where $N_{\text{count}}$ represents the number of vehicles counted by the model.

**Table 1  Prediction results under sunny environment.**

| Vehicle type | $N_{true}$ | $N_{count}$ | Accuracy (%) |
|---|---|---|---|
| SUV | 35 | 34 | 97.14 |
| Sedan | 83 | 78 | 93.98 |
| Taxi | 14 | 13 | 92.85 |
| Van | 19 | 23 | 78.95 |
| Truck-Box | 1 | 1 | 100 |
| Bus | 0 | 0 | – |
| Mini Van | 1 | 1 | 100 |
| Truck-Util | 0 | 0 | – |
| Pickup | 0 | 0 | – |
| Total | 153 | 150 | 98.00 |

## RESULTS AND ANALYSIS

Then the prediction model introduced in this article was utilized to test the performance of the traffic data under four different conditions, namely sunny day, cloudy day, rainy day and evening, and the test results are illustrated in Tables 1, 2, 3 and 4, respectively.

Observing the prediction results in Tables 1–4, we can get the following conclusions.

(1) In general, the traffic flow prediction algorithm based on the GRU model designed in this article achieves more satisfactory traffic flow calculation results in four conditions: sunny day, cloudy day, rainy day and evening, and the prediction accuracy of the model is relatively high. This may be due to the fact that for the traffic flow data characteristics, this article first utilizes GRU model to build a new deep learning neural network model to directly process the original data, and then fuses the fine-grained traffic flow statistical calculation method to obtain the prediction results, which greatly improves the robustness and generalization ability of the algorithm.

(2) Secondly, in actual daily life, whether in sunny, cloudy, rainy days or evenings, Sedan, SUV, Taxi, and Bus are the general-purpose transportation vehicles, and the probability of appearing is much higher than that of Pickup and Truck-Util. This phenomenon is also supported by the results of the above four cases of traffic flow calculation for different vehicle types. This is also very consistent with the actual situation.

(3) Finally, the above simulation experiments also fully demonstrate the effectiveness of the prediction model designed in this article in the IoV environment. Through further optimization in the future, we hope to provide users with more reliable traffic flow change trends, so as to provide people with reasonable travel arrangements in advance, avoid traffic jams and optimize road network traffic.

## CONCLUSION

As China's automobile industry grows rapidly, the number of automobiles has exploded, causing many traffic problems. The emergence of IoV technology provides a new technical means to alleviate the current traffic pressure. Relying on IoV, while combining advanced neural network technology, traffic optimization is realized, thus improving people's travel

**Table 2 Prediction results under cloudy environment.**

| Vehicle type | $N_{true}$ | $N_{count}$ | Accuracy (%) |
|---|---|---|---|
| SUV | 10 | 11 | 90 |
| Sedan | 31 | 32 | 96.77 |
| Taxi | 11 | 9 | 81.82 |
| Van | 1 | 1 | 100 |
| Truck-Box | 2 | 1 | 50 |
| Bus | 0 | 0 | – |
| Mini Van | 0 | 0 | – |
| Truck-Util | 1 | 1 | 100 |
| Pickup | 0 | 0 | – |
| Total | 56 | 55 | 98.21 |

**Table 3 Prediction results under rainy environment.**

| Vehicle type | $N_{true}$ | $N_{count}$ | Accuracy (%) |
|---|---|---|---|
| SUV | 6 | 5 | 83.33 |
| Sedan | 15 | 17 | 86.67 |
| Taxi | 0 | 0 | – |
| Van | 4 | 5 | 75 |
| Truck-Box | 4 | 2 | 50 |
| Bus | 1 | 1 | 100 |
| Mini Van | 1 | 0 | 0 |
| Truck-Util | 1 | 0 | 0 |
| Pickup | 0 | 0 | – |
| Total | 32 | 30 | 93.33 |

**Table 4 Prediction results in the evening environment.**

| Vehicle type | $N_{true}$ | $N_{count}$ | Accuracy (%) |
|---|---|---|---|
| SUV | 17 | 16 | 94.12 |
| Sedan | 41 | 40 | 97.56 |
| Taxi | 18 | 20 | 88.89 |
| Van | 5 | 4 | 80 |
| Truck-Box | 0 | 0 | – |
| Bus | 0 | 0 | – |
| Mini Van | 1 | 1 | 100 |
| Truck-Util | 0 | 0 | – |
| Pickup | 1 | 0 | 0 |
| Total | 83 | 81 | 97.53 |

efficiency and optimizing their lifestyles. In view of this, this article focuses on traffic flow prediction algorithm, and designs and implements a deep neural network with high accuracy and generalization capability based on real-time traffic data collected in the IoV environment to achieve analysis and prediction of short-time traffic flow. On the one

hand, the research suggests using the GRU model to achieve the prediction of the trend of traffic flow in the near future in order to address the issues of incomplete data feature extraction and low prediction accuracy in the existing traffic flow prediction algorithms. On the other hand, a fine-grained traffic flow statistical calculation method is suggested for the IoV scenario for the constructed deep neural network prediction model in order to realize the traffic information statistics of various vehicle types. The simulation results demonstrate that the new algorithm has good performance in traffic flow statistics in different scenarios. Therefore, it helps to provide reference for people' travel arrangement, so as to avoid congested roads, improve people's travel efficiency and also help to relieve traffic pressure. Although the research in this article has achieved some phased results, there are still some shortcomings to be further improved. Future research work mainly includes: First, more optimization is required for the traffic flow prediction algorithm provided in this article. The current algorithm only measures the traffic flow as an element. However, considering the complexity of factors affecting traffic flow, this article will further use feature engineering technology to mine more correlation factors and construct a reasonable weight matrix. Then the GRU network model will be used for feature training to increase the accuracy of multi-step prediction. Secondly, in the process of traffic flow prediction, the current algorithm only performs statistical analysis for traffic changes within a specific region. Therefore, future research will optimizes the prediction model to enhance its ability to handle more complex real-world scenarios.

### Funding
This work was supported by the Science and Technology Project of Chongqing Education Commission—Research on the Design and Application of Digital Twin Biochemistry for Smart Agricultural Production Monitoring System under the Background of Rural Revitalization (Fund No.: KJQN202204009). The funders had no role in study design, data collection and analysis, decision to publish, or preparation of the manuscript.

### Grant Disclosures
The following grant information was disclosed by the authors:
Science and Technology Project of Chongqing Education Commission—Research on the Design and Application of Digital Twin Biochemistry for Smart Agricultural Production Monitoring System under the Background of Rural Revitalization: KJQN202204009.

### Competing Interests
The authors declare there are no competing interests.

### Author Contributions
- Wu Wen conceived and designed the experiments, performed the experiments, analyzed the data, performed the computation work, prepared figures and/or tables, authored or reviewed drafts of the article, and approved the final draft.

- Dongliang Xu performed the experiments, analyzed the data, performed the computation work, prepared figures and/or tables, authored or reviewed drafts of the article, and approved the final draft.
- Yang Xia performed the experiments, analyzed the data, performed the computation work, prepared figures and/or tables, authored or reviewed drafts of the article, and approved the final draft.

### Data Availability

The data is available at DETRAC-Train-Images: https://detrac-db.rit.albany.edu/download.

The raw measurements and model are available in the Supplemental Files.

### Supplemental Information

Supplemental information for this article can be found online at http://dx.doi.org/10.7717/peerj-cs.1411#supplemental-information.

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
