# Peer review of "A novel traffic optimization method using GRU based deep neural network for the IoV system"

_PeerJ Computer Science, doi:10.7717/peerj-cs.1411_

## Round 0.1 · original submission · Major Revisions

· Academic Editor

Major Revisions

According to the comments from two reviewers, I suggest a major revision. Please carefully revise the manuscript and resubmit it for a second review.

Reviewer 1 ·

Basic reporting

This article focuses on the traffic flow prediction algorithm and designs and implements a deep neural network with high accuracy and the capability to generalize based on real-time traffic data collected in the IoV environment to achieve analysis and prediction of short-time traffic flow. Specifically, the paper analyzes and predicts the flow of traffic in the IoV environment.

Experimental design

Please provide a more in-depth explanation of the experimental results that are presented in Tables 1–4.

Validity of the findings

As a result, it is useful to provide reference for people's travel arrangements, which helps people avoid congested roads, improves people's travel efficiency, and also contributes to the relief of traffic pressure. Therefore, the research work that was done for this study has a very real significance, and I believe that it is acceptable to acknowledge this. In addition to that, a few adjustments are essential.

Additional comments

This article focuses on the traffic flow prediction algorithm and designs and implements a deep neural network with high accuracy and the capability to generalize based on real-time traffic data collected in the IoV environment to achieve analysis and prediction of short-time traffic flow. Specifically, the paper analyzes and predicts the flow of traffic in the IoV environment. As a result, it is useful to provide reference for people's travel arrangements, which helps people avoid congested roads, improves people's travel efficiency, and also contributes to the relief of traffic pressure. Therefore, the research work that was done for this study has a very real significance, and I believe that it is acceptable to acknowledge this. In addition to that, a few adjustments are essential. The following are some comments and suggestions I have regarding the paper:

1. It is mentioned in the "abstract" that "a fine-grained traffic flow statistics approach is suggest in the paper." The "paper" refers to the "abstract." It is imperative, in my opinion, for the developer of the algorithm to provide some additional details regarding its operation.
2. In the second paragraph of the "Introduction," the primary focus is placed on introducing the research work that is associated with the Internet of Vehicles and traffic flow. If the author is able to elaborate on the nature of the connection between the two, I would greatly appreciate it.
3. The sigmoid and tanh activation functions are described in the article as being the most typical choices for the activation function f, which is depicted in Figure 2 and appears in the formula (1). In order to provide a reference for other research, I believe it is necessary to provide a more in-depth explanation of the characteristics shared by these two functions.
4. I believe it is necessary for the author to explain the operating principle of error back propagation for Figure 4.
5. The statement can be found in Section 3.1 of the paper and states, "The GRU neural network is a model improved on the basis of LSTM." Kindly explain the difference between the two models so that the audience can comprehend the rationale behind the selection of the GRU model.
6. Regarding Figure 7, I am hoping that the author will provide some sort of explanation regarding the statistical process involved in the fine-grained traffic flow method.
7. Could you please explain what each of these formula variables means?
8. Please provide a more in-depth explanation of the experimental results that are presented in Tables 1–4.

Reviewer 2 ·

Basic reporting

In order to better improve road network traffic conditions in an IoV environment, this research focuses on the traffic flow prediction method that is based on deep learning to boost traffic efficiency and safety. This will make it possible to further enhance the efficiency of the road system's traffic management systems.

Experimental design

The simulation results show that the new method provides reliable traffic flow data under a wide range of conditions.

Validity of the findings

To begin, the research team employed a deep learning-based GRU model for short-term traffic flow prediction in order to address the issues that existing algorithms lack the capacity to automatically extract data features and that the model expression power is restricted. Second, a method for calculating fine-grained traffic flow statistics that is suitable for the IoV scenario is proposed in line with the results of the constructed deep learning model.

Additional comments

For this reason, I argue that the work should be approved after a few tweaks have been made. The manuscript has to be revised, so please do that. See the comments here.
1)At the conclusion of the section titled <Introduction,> the body of the paper provides a synopsis of the pertinent research and an overview of the primary research contents of this paper. However, I believe that a summary of the research work that was done in this paper is required, and I ask that the content of this section be refined.
2)The fundamentals of deep learning as well as its practical applications are discussed in sub-section 2.1. I really hope that the author can provide a more in-depth explanation of the benefits of this method so that other researchers can have a better understanding of the reasoning behind the selection of the deep learning model for this paper.
3)The input layer, hidden layer, and output layer of the deep feedforward network are each introduced in the paper for Fig. 3, and the functions of the input layer and the output layer are explained. I think it would be helpful if the author provided some additional context regarding the function of the hidden layer.
4)For the Eq. (9), could you please provide a comprehensive explanation of what it means?
5)In the section of the paper devoted to the experiments, I would like the authors to provide a more in-depth description of the material covered in the section titled <Experimental Design.>
6)The phrase <the traffic dataset in this experiment is mainly derived from a hybrid dataset composed of manually labeled dataset and UA-DETRAC public dataset> can be found in sub-section 4.1 of the article. Kindly explain the reason why.
7)In the part of the paper titled <Conclusion,> the author provides a synopsis of the research that was conducted for the paper. Nevertheless, the flaws and additional research that needs to be done have not been mentioned, and I am hoping that the author will add to it and make it better.

---

## Round 0.2 · accepted · Accept

· Academic Editor

Accept

Two reviewers give positive comments . Therefore, I recommend acceptance.

Reviewer 1 ·

Basic reporting

no comment

Experimental design

no comment

Validity of the findings

no comment

Additional comments

The authors have revised this paper and addressed my main concerns. I am satisfied with the updates. I recommend that this paper can be accepted.

Reviewer 2 ·

Basic reporting

Urban traffic pressure is increasingly prominent, and the situation of traffic congestion and traffic safety is not optimistic. In this context, Internet of Vehicles( also known as "IoV") opens up a new way to relieve urban traffic pressure. Therefore, in order to further optimize the road network traffic conditions in the IoV environment, this research focuses on the traffic flow prediction algorithm on the basis of deep learning to enhance traffic efficiency and safety. The study investigates the short-time traffic flow prediction by combining the characteristics of the IoV environment.

Experimental design

The IoV environment is constructed to confirm the effectiveness of the prediction model. The prediction results prove that the new algorithm has good performance in traffic flow statistics in different scenarios.

Validity of the findings

A fine-grained traffic flow statistics approach suitable for the IoV situation is suggested in accordance with the deep learning model that was built. The algorithm sends the vehicle characteristic data obtained through GRU model training into the fine-grained traffic flow statistics algorithm, so as to realize the statistics of traffic information of various types of vehicles. The advantage of this algorithm is that it can well count the traffic flow of multiple lanes, so as to better predict the current traffic status and achieve traffic optimization.

Additional comments

Current version of the manuscript can be accepted.